# Syndecan-1 Facilitates the Human Mesenchymal Stem Cell Osteo-Adipogenic Balance

**DOI:** 10.3390/ijms21113884

**Published:** 2020-05-29

**Authors:** Chieh Yu, Ian W. Peall, Son H. Pham, Rachel K. Okolicsanyi, Lyn R. Griffiths, Larisa M. Haupt

**Affiliations:** Queensland University of Technology, Genomics Research Centre, School of Biomedical Sciences, IHBI, Brisbane, QLD 4059, Australia; chieh.yu@ucsf.edu (C.Y.); ian.peall@hdr.qut.edu.au (I.W.P.); hoangson.pham@hdr.qut.edu.au (S.H.P.); r.okolicsanyi@qut.edu.au (R.K.O.); lyn.griffiths@qut.edu.au (L.R.G.)

**Keywords:** mesenchymal stem cells, heparan sulfate proteoglycans, syndecan-1, osteogenesis, adipogenesis, lineage fate

## Abstract

Bone marrow-derived human mesenchymal stems cells (hMSCs) are precursors to adipocyte and osteoblast lineage cells. Dysregulation of the osteo-adipogenic balance has been implicated in pathological conditions involving bone loss. Heparan sulfate proteoglycans (HSPGs) such as cell membrane-bound syndecans (SDCs) and glypicans (GPCs) mediate hMSC lineage differentiation and with syndecan-1 (SDC-1) reported in both adipogenesis and osteogenesis, these macromolecules are potential regulators of the osteo-adipogenic balance. Here, we disrupted the HSPG profile in primary hMSC cultures via temporal knockdown (KD) of SDC-1 using RNA interference (RNAi) in undifferentiated, osteogenic and adipogenic differentiated hMSCs. SDC-1 KD cultures were examined for osteogenic and adipogenic lineage markers along with changes in HSPG profile and common signalling pathways implicated in hMSC lineage fate. Undifferentiated hMSC SDC-1 KD cultures exhibited a pro-adipogenic phenotype with subsequent osteogenic differentiation demonstrating enhanced maturation of osteoblasts. In cultures where SDC-1 KD was performed following initiation of differentiation, increased adipogenic gene and protein marker expression along with increased Oil Red O staining identified enhanced adipogenesis, with impaired osteogenesis also observed in these cultures. These findings implicate SDC-1 as a facilitator of the hMSC osteo-adipogenic balance during early induction of lineage differentiation.

## 1. Introduction

Mesenchymal stem cells (MSCs) are adult stem cells known for their ability to self-renew and differentiate into multiple mesenchymal cell types, including osteoblasts, adipocytes, chondrocytes, fibroblasts and smooth muscle cells [1,2]. As the common precursor to both osteoblasts and adipocytes, the MSC osteo-adipogenic balance is delicately and tightly regulated. Dysregulation has been implicated in multiple pathological conditions, including bone diseases (osteoporosis, osteopenia), obesity and diabetes [3,4,5,6]. The relationship between osteogenic and adipogenic lineage differentiation of MSCs is mutually inhibitory, such that osteogenic commitment blocks adipogenic differentiation and vice versa [7]. This has been reported in animal studies of ageing and stress conditions of the bone marrow environment, such as via chemotherapy and irradiation. Under these stressors, decreased numbers of osteogenic precursors and increased marrow adiposity have been observed [8,9,10]. Additionally, this observation was also seen in clinical patients through imaging studies, demonstrating similarities and relevance to human conditions [11,12,13]. Therefore, the identification of the regulators of MSC differentiation is critical to our ability to understand and repair damage due to disease and conditions relating to an osteo-adipogenic imbalance in the bone marrow.

The stem cell niche, the microenvironment that encompasses all of the elements immediately surrounding stem cells when they are in their naïve state, provides a wide range of intrinsic and extrinsic factors that influence MSC fate or lineage specificity [14]. Molecular regulation of MSC osteo-adipogenic differentiation involves a complex interplay between a plethora of growth factors, their associated signalling pathways and direct or indirect downstream targets throughout different stages of development [3,15]. Differentiation programs are activated when transcriptional factors are induced. Adipogenesis is first initiated by the activation of CCAAT/enhancer-binding protein β (C/EBPβ) and C/EBPδ which then triggers the ‘master’ adipogenic transcription factors, C/EBPα and peroxisome proliferator-activated receptor γ (PPARγ) [16,17]. Osteogenic differentiation is mediated by expression of the runt-related transcription factor 2 (RUNX2), considered the master regulator, controlling osteoblast differentiation, gene expression and function [18].

The ECM is a major component of the localised niche influencing lineage specific differentiation [19]. The heparan sulfate proteoglycans (HSPGs) are ubiquitously present in the ECM and have been established as key mediators of numerous cellular processes, including proliferation, self-renewal, lineage commitment and differentiation. HSPGs consist of a core protein to which one or more heparan sulfate (HS) glycosaminoglycan (GAG) chains are attached. They can be matrix-localised or cell surface bound (including the transmembrane syndecans (SDCs) and glycosylphosphatidylinositol (GPI)-anchored glypicans (GPCs)) [20]. HSPG core proteins are encoded by genes and transported to the Golgi for an elaborate and highly regulated posttranslational process of HS GAG biosynthesis orchestrated by a suite of biosynthetic enzymes. The combination of core proteins, HS GAG chain length, number of chains, and sulfation pattern gives rise to a complex protein with highly variable structure and function, allowing interactions with ECM components including structural proteins, cytokines, and growth factors such as fibroblast growth factors (FGFs), bone morphogenetic proteins (BMPs) and Wnts to exert their functions [21,22,23]. As such, cells within the local microenvironment influence the configuration of HSPG protein-binding sites on their GAG chains throughout all stages of development [24].

In conjunction with the roles of HS GAGs, the controlled expression of specific HSPG core proteins has been shown to be crucial in osteogenic and adipogenic differentiation programs. We have previously demonstrated the contribution of the HSPG glypican-3 (GPC-3) to MC3T3-E1 pre-osteoblast cell osteogenesis through interactions with RUNX2 to promote differentiation [25]. The additional involvement of other HSPGs including GPC-2 and the SDCs, in particular SDC-1, is also evident through their upregulation in gene expression [25]. SDCs-1-4 regulate the osteoblast lineage through interactions with FGFs and Wnts [26,27,28,29]. Interestingly, only SDC-1 and SDC-4 have been reported in adipogenesis with Zaragosi et al. (2014) demonstrating that SDC-1 also promotes proliferation of pre-adipocytes and inhibits their differentiation potential [30], while Landry and colleagues reported that SDC-4 is transiently expressed throughout adipocyte differentiation [31]. Thus, further studies of HSPGs, in particular the SDCs, in the context of hMSC osteo-adipogenic balance would provide greater understanding of initial MSC fate regulators and identify potential targets to facilitate or control lineage specification.

In this study, we examined the role of the HSPG core protein SDC-1 in regulating the osteo-adipogenic balance in hMSCs. With SDC-1 implicated in both osteogenesis and adipogenesis, it may provide a potential temporal biomarker and a novel clinical target for managing bone marrow adiposity and bone loss.

## 2. Results

SDC-1 expression was downregulated in bone marrow-derived hMSC populations (*n* = 2) through SDC-1 specific siRNA knockdown (KD) in undifferentiated hMSC cultures, as well as during hMSC osteogenic (OS) and adipogenic (AD) differentiation cultures. SDC-1 specific siRNA was incubated with cells for 72 h or 96 h for RNA or protein knockdown, respectively. For osteogenic differentiation, hMSCs were cultured for 21 days in Osteogenic Induction Medium (OIM). For adipogenic differentiation, hMSC KD cultures then underwent cycles of Adipogenesis Induction Medium (AIM) and Adipogenesis Maintenance Medium (AMM) for a total of 22 days. In hMSC differentiation cultures, SDC-1 KD was performed after initiation of differentiation, with the cultures incubated with SDC-1 specific siRNA for 96 h (during day 2-day 6 of differentiation, to ensure protein KD) then the siRNA was removed through media change and cultures continued until terminal differentiation. The effect of SDC-1 KD on undifferentiated and differentiated hMSC culture proliferation, viability, HSPG profile, osteo-adipogenic lineage marker expression and functionality was then examined.

### 2.1. Disrupted HSPG Profile Impedes Undifferentiated hMSC Proliferation

The effect of SDC-1 KD on undifferentiated hMSC proliferation and viability was examined after 72 h of incubation with SDC-1 siRNA. With the initial seeding density of 4.5 × 10^4^ cells/condition (triplicates pooled), the control (CTRL) cultures, comprised of untreated (UT) and non-targeting/scrambled (SCR) siRNA treated cultures, underwent one population doubling by the completion of SDC-1 KD with an average cell count of 7.0 × 10^4^ cells after 72 h (Figure 1A). In contrast, proliferation in SDC-1 KD cultures was observed to be minimal, with a cell count of 5.0 × 10^4^ cells following 72 h in culture, with 48% fewer cells than in CTRL cultures (Figure 1A). However, similarly high cell viability in the cultures were recorded, with CTRL cultures maintaining 83% viability and SDC-1 KD cultures 90% viability, detected by trypan blue exclusion and supported by FDA/PI cell staining (Figure 1A,B).

The effect of SDC-1 KD was then examined in hMSC osteogenic and adipogenic lineages, with osteogenic or adipogenic differentiation initiated in cultures via the addition of lineage specific culture media. No significant effect on cell number was observed in hMSC between CTRL_OS_ (osteogenic control) and SDC-1 KD_OS_ (osteogenic SDC-1 KD) cultures (Figure 1A), whereas SDC-1 KD_AD_ (SDC-1 KD adipogenic) cultures were found to have significantly higher average cell numbers (*p* < 0.05) when compared to CTRL_AD_ (adipogenic control) cultures (Figure 1D). High cell viability (>90%) was recorded for CTRL_AD/OS_ and SDC-1 KD_AD/OS_ cultures undergoing osteogenesis and adipogenesis, when assessed by trypan blue exclusion (Figure 1C,D) and FDA/PI staining (Figure 1E).

### 2.2. Expression of HS Biosynthetic Machinery and HSPG Core Protein in Niche-Altered hMSC Cultures

Treatment of undifferentiated hMSC cultures with SDC-1-specific siRNA resulted in a 20% downregulation of SDC-1 expression at the transcript level when compared to CTRL cultures (Figure 2A). Interestingly, at the protein level, Western blot (WB) analysis identified an 18% upregulation of SDC-1 in these cultures (Figure 2B). In these cultures, Q-PCR analysis showed downregulation of SDC-1 resulted in the upregulation of the majority of the other HSPG core proteins, including a significant increase in the gene expression level of GPC-2 (*p* = 0.0424), GPC-3 (*p* = 0.0167) and GPC-4 (*p* = 0.0481) (Appendix A).

As hMSC cultures underwent osteogenic and adipogenic lineage specific differentiation following SDC-1 KD during early differentiation, examination of HSPG core protein gene expression showed no significant changes in SDCs (Figure 2C) or GPCs (Figure 2D) by Q-PCR in both lineage specific culture conditions. However, GPC-2 expression was shown to be diminished in SDC-1 KD_OS_ cultures, whereas GPC-3 expression was diminished in SDC-1 KD_AD_ cultures (Figure 2D). Protein analysis of HSPG core proteins using the pan-HS 3G10 antibody revealed upregulation of SDC-1 (32 kDa), SDC-3 (46 kDa), and a 25 kDa size HS-carrying protein in the SDC-1 KD_AD_ cultures (Figure 2E). The increased SDC-1 protein expression correlated with the upregulation observed at the transcript level, indicating SDC-1 KD was not sustained throughout lineage differentiation.

Following SDC-1 KD, widespread downregulation of HS biosynthesis and modification enzyme gene expression was observed in undifferentiated hMSCs, with the exception of HS 6-*O*-sulfotransferase isoform 1 (HS6ST1) (Appendix A). HS6ST1 performs 6-*O*-sulfation of heparan sulfate and is the only enzyme that displayed increased gene expression following SDC-1 KD. A non-significant downregulation of the two exostosin genes (EXT1-2), C5-epimerase (C5-EP) (Appendix A), *N*-deacetylase/*N*-sulfotransferase 1-2 (NDST1-2) (Appendix A), 6-*O*-sulfatases 1-2 (SULF1-2), and heparanase (HPSE) (Appendix A) was also observed in the KD cultures. These enzymes perform HS chain elongation, epimerisation, *N*-sulfation, 6-*O*-desulfation and HS cleavage, respectively. Significant downregulation of gene expression was observed for NDST1 (*p* = 0.0006; Appendix A), a crucial enzyme for *N*-sulfation and the overall design of HS chains, as well as SULF1 (*p* = 0.0455; Appendix A), which removes 6-*O*-sulfation [32].

In the hMSC SDC-1 KD_OS_ lineage differentiated cultures, significant downregulation was observed for EXT1 (*p* = 2.022 × 10^−6^), EXT2 (*p* = 4.061 × 10^−6^) and C5-EP (*p* = 1.275 × 10^−6^) gene expression (Figure 3A). Similarly, a significant downregulation of EXT1 (*p* = 4.015 × 10^−5^), EXT2 (*p* = 2.124 × 10^−5^), and C5-EP (*p* = 2.129 × 10^−6^) was also observed in the hMSC SDC-1 KD_AD_ cultures (Figure 3A). While no significant changes in NDST enzyme expression levels were observed in the SDC-1 KD_OS_ cultures, significant upregulation of NDST1 (*p* = 0.0124) was observed in the SDC-1 KD_AD_ cells along with a non-significant upregulation of NDST2 gene expression (Figure 3B). Similarly, non-significant changes for *O*-sulfation enzymes were observed in the SDC-1 KD_OS_ cultures with significant upregulation observed in the SDC-1 KD_AD_ cultures for both HS2ST1 (*p* = 0.0055) and HS6ST1 (*p* = 0.0054) (Figure 3C). Similar gene expression levels of HS modification enzymes SULF1, SULF2 and HPSE were observed in CTRL_AD/OS_ and SDC-1 KD_AD/OS_ conditions in both lineage specific cultures (Figure 3D).

### 2.3. SDC-1 KD Led to the Downregulation of Osteogenic Markers and up-Regulation of Adipogenic Markers

To examine the effect of HSPG disruption by SDC-1 KD on the hMSC osteo-adipogenic balance, selected osteogenic and adipogenic lineage markers were then examined by Q-PCR and WB. In undifferentiated hMSC SDC-1 KD cultures, the majority of osteogenic markers examined were observed to be downregulated (Appendix A), including the master transcription factor runt-related transcription factor 2 (RUNX2), collagen, type I, alpha 1 (COL1A1) and osteocalcin (OCN). In addition, COL1A1 was shown to be downregulated at the protein level by 61% in SDC-1 KD cultures when compared to CTRL cultures, suggesting decreased osteogenic potential (Appendix A). Of the adipogenic markers examined, upregulation of adiponectin (ADIPO-Q; *p* = 0.0363), PPARγ1 and C/EBPδ was observed in undifferentiated SDC-1 KD cultures (Appendix A), indicating pro-adipogenic potential of the cultures.

Expression of these osteogenic and adipogenic lineage markers were then examined in hMSC SDC-1 KD_AD/OS_ differentiated cultures. The osteogenic markers RUNX2, COL1A1, and alkaline phosphatase (AP) in SDC-1 KD_OS_ cultures showed higher gene expression levels when compared to CTRL_OS_ cultures (Figure 4A), although this was not found to be significant. However, COL1A1 protein was found to be reduced in the SDC-1 KD_OS_ culture by 45% in comparison to the CTRL_OS_ cultures (Figure 4B). In the SDC-1 KD_AD_ cultures, RUNX2 was identified to be significantly upregulated in comparison to the CTRL_AD_ adipogenic cultures (*p* = 1.099 × 10^−5^), with AP significantly downregulated following SDC-1 KD (*p* = 0.0269), and no significant changes were observed between CTRL_AD/OS_ and SDC-1 KD_AD/OS_ cultures for COL1A1 and OCN (Figure 4A). Western blot analysis of COL1A1 in SDC-1 KD_AD_ cultures also showed a 53% reduction in protein when compared to the CTRL_AD_ cultures (Figure 4C).

Examination of selected adipogenic markers in SDC-1 KD_AD/OS_ lineage specific cultures showed the majority of the adipogenic markers to be non-significantly upregulated in the SDC-1 KD_OS_ cultures, in comparison to CTRL_OS_ cultures. With the exception of PPARγ2, higher gene expression levels of C/EBPα, C/EBPδ, PPARγ1, and ADIPO-Q were observed in SDC-1 KD_OS_ cultures (Figure 4D). In the SDC-1 KD_AD_ cultures, gene expression levels of C/EBPα, C/EBPδ, PPARγ2, and ADIPO-Q remained unchanged between CTRL_AD_ and SDC-1 KD_AD_ cultures, although PPARγ1 was observed to be significantly downregulated (*p* = 0.0354) (Figure 4D). Western blot analysis supported these observed gene expression changes, with PPARγ protein expression identified to be similar in the CTRL_AD_ and SDC-1 KD_AD_ conditions (Figure 4E).

### 2.4. BMP2 Signalling Was Induced Temporally Following SDC-1 KD in Undifferentiated hMSC

Indication of an osteo-adipogenic imbalance, along with an observed pro-adipogenic potential following SDC-1 KD, triggered investigation of common signalling pathways implicated in osteogenic and adipogenic differentiation. Representative of the BMP signalling pathway, endogenous BMP2 ligand transcript levels were found to be increased in SDC-1 KD cultures (Appendix A), indicating activation of the BMP2 signalling pathway in undifferentiated hMSC SDC-1 KD cultures. BMP4 gene expression was reduced in SDC-1 KD (Appendix A) cultures while BMP7 expression was undetected (data not shown). In addition, gene expression of BMP receptor IA (BMPR-IA) was found to be upregulated in SCD-1 KD cultures, suggesting it as the more active receptor, with BMPR-IB observed to be downregulated (Appendix A).

FGF signalling was also examined through gene expression of the ligand FGF2 and the receptors FGFR1-4. Gene expression levels of FGF2, FGFR1 and FGFR2 were identified to be non-significantly downregulated in undifferentiated hMSC SDC-1 KD cultures (Appendix A), with FGFR3 and FGFR4 expression undetected in either CTRL or SDC-1 KD cultures (data not shown). Similarly, gene expression of Wnt3a and the receptor Frizzled-1 (FZD1) were also shown to be non-significantly downregulated in SDC-1 KD cultures, indicating a decreased requirement for canonical Wnt signalling (Appendix A). Interestingly, when sonic hedgehog (SHH) signalling was examined, gene expression of both the ligand SHH and the G protein-coupled receptor-like receptor Smoothened (SMO) were observed to be downregulated, whereas the canonical receptor Patched-1 (PTCH1) was upregulated in the undifferentiated hMSC SDC-1 KD cultures (Appendix A).

However, following hMSC lineage-specific differentiation in SDC-1 KD_AD/OS_ conditions, no significant changes were observed in gene expression levels of these pathway components between CTRL_AD/OS_ and SDC-1 KD_AD/OS_ terminal cultures (Figure 5). In SDC-1 KD_AD_ cultures, a non-significant decreased expression of BMP ligands, including BMP2 and BMP4 (Figure 5A), and FGF pathway components FGF2 and FGFR1 and FGFR2 (Figure 5B) was observed with FGFR3 and FGFR4 undetected in both SDC-1 KD_AD/OD_ culture conditions (data not shown). In the SDC-1 KD_OS_ cultures, FGFR2 expression was reduced, but this was not found to be significant (Figure 5B). No significant differences in Wnt (Figure 5C) and SHH (Figure 5D) signalling expression was observed between CTRL_AD/OS_ and SDC-1 KD_AD/OS_ conditions in the osteogenic and adipogenic cultures.

### 2.5. Enhanced Adipogenesis and Impaired Osteoblast Maturation Was Observed in hMSC SDC-1 KD_AD/OS_ Cultures

To further examine the role of SDC-1 in the hMSC osteo-adipogenic balance, hMSC SDC-1 KD_AD/OS_ cultures were then examined for their capacity to complete their differentiation programs. Terminal adipogenic cultures were stained with Oil Red O to examine liquid vacuole formation. Alizarin Red and von Kossa staining was performed at the completion of 21 days of osteogenic differentiation to examine calcification and mineralisation, respectively. The ability to calcify and mineralise are features of fully mature osteoblasts [33].

In SDC-1 KD_AD/OS_ conditions, adipogenesis was observed to be enhanced while osteogenesis was impaired (representative images shown in Figure 6; stained cultures shown in Appendix A). Oil Red O staining was greater in SDC-1 KD_AD_ conditions when compared to CTRL_AD_ cultures (Figure 6A). Quantitation of optical density (OD) of the stained cultures (normalised to cell number due to the observed differences in cell numbers between culture conditions (Figure 1C,D), showed the SDC-1 KD_AD_ cultures to have a 20% increase in Oil Red O staining (Figure 6A). In the SDC-1 KD_OS_ cultures, calcium deposits in terminally differentiated osteogenic cultures stained for Alizarin Red, showed a 58% reduction of staining in the SDC-1 KD_OS_ than CTRL_OS_ cultures (Figure 6B). Von Kossa staining of calcium salt anions [34] identified no difference between CTRL_OS_ and SDC-1 KD_OS_ cultures (Figure 6B).

Interestingly, where osteogenic differentiation was initiated following SDC-1 KD in hMSC cultures, improved osteogenesis was observed when compared to CTRL cultures (Appendix A). This was despite the apparent pro-adipogenic phenotype observed in undifferentiated SDC-1 KD hMSCs. OD quantitation and normalisation to the cell number in SDC-1 KD cultures (due to differences in cell numbers observed between culture conditions, Figure 1A) identified higher calcification (+49% Alizarin Red) and mineralisation (+57% von Kossa) (Appendix A), indicating greater osteoblast differentiation and maturation.

## 3. Discussion

A deeper understanding of MSC lineage fate regulators will enable greater control over the use of hMSCs in regenerative medicine applications. MSCs possess numerous properties such as their ease of isolation, high ex vivo expansion capability, homing ability, immunomodulatory properties and multipotency, making them an ideal cell type for use in tissue repair and regeneration [35,36]. In vivo, MSCs function as precursors to both osteoblasts and adipocytes [3], making their regulation essential for maintaining the bone marrow osteo-adipogenic balance. As such, a better understanding of the inverse relationship of MSC osteoblast and adipocyte lineages has significant implications, from insights into pathophysiological conditions such as osteoporosis and obesity, to the development of treatments for these disorders and MSC engineering for bone tissue repair [3,37].

HSPGs located on cell surfaces or within the ECM have a vast range of functions due to their highly diverse structures. The combination of core proteins, variable HS side chain length, chain number, sulfation patterns and constant remodelling contribute to their various functions, including cell proliferation, differentiation and maintenance [21]. In this study, we investigated the involvement of HSPGs in regulating the osteo-adipogenic balance using an in vitro SDC-1 KD hMSC model. We targeted SDC-1 due to its previously reported role in both osteogenesis and adipogenesis [26,29,30,38]. Specifically, SDC-1 gene expression was previously demonstrated to be downregulated during human adipocyte differentiation [30,39], while murine studies showed SDC-1 was significantly upregulated in osteoprogenitors and osteogenic differentiation [25,29].

As SDC-1 KD efficiency was only examined at one timepoint following KD, at 72 h for examination of RNA and 96 h for protein expression, this likely explains why SDC-1 KD was only achieved at the transcript level and not also at the protein level, although a delayed downregulation of SDC-1 protein may have occurred. Another possibility for this observed increase in protein is the shedding of SDC-1 from the cell surface. In this process, the ectodomain of SDC-1 is subjected to proteolytic cleavage in response to physiological agents, releasing the protein into the extracellular matrix as a soluble ligand [40]. As Western blotting was performed using cell lysates, containing both cell surface-bound and ECM-localised HSPGs, while SDC-1 was shown to be downregulated due to decreased transcription following KD, SDC-1 shedding may have contributed to the slight increase in total SDC-1 protein level observed at the 96 h timepoint.

SDC-1 is associated with proliferation through its interaction with growth factors and is often upregulated in cancer cells during tumour progression [41,42]. In this study, SDC-1 KD resulted in a 20% decrease in SDC-1 gene expression, which is considered to be a knockdown of low efficiency. Primary cells, including hMSCs, are generally more difficult to transfect than cell lines [43]. However, the 20% decrease in SDC-1 gene expression resulted in a 48% reduction in cell number in undifferentiated SDC-1 KD cultures when compared to undifferentiated CTRL cultures (Figure 1A). In a previous study in human multipotent adipose stem (hMADS) cells, similar to hMSCs in that they are derived from the mesenchyme, they were observed to have an approximate 50% reduction in cell number following SDC-1 KD [30]. The decreased cell number observed in this study following SDC-1 KD indicates an effect on SDC-1 function. The significant changes observed in HSPG core protein (Appendix A) and biosynthetic enzyme expression (Appendix A) also indicate the HSPG profile of hMSC cultures was altered following SDC-1 KD. This is further supported by the observed downregulation of osteogenic markers at the gene and protein level (Appendix A), and the significant upregulation of the adipogenic marker (ADIPO-Q; Appendix A), suggesting an altered osteo-adipogenic balance in the hMSC SDC-1 KD cultures.

Undifferentiated hMSC cultures exhibited a pro-adipogenic phenotype following SDC-1 KD, as is evident by the significant upregulation of ADIPO-Q expression, along with key transcription factors PPARγ1 and C/EBPδ, and the accompanying downregulation of the osteogenic markers RUNX2 and COL1A1, a downstream target of RUNX2 [18], determined by gene and protein expression analyses. Interestingly, the induction of BMP2 and PTCH expression in undifferentiated SDC-1 KD cultures suggests the BMP and SHH signalling pathways are activated. BMP2 is a predominantly pro-osteogenic factor for hMSCs [44,45] and SHH signalling is also required for hMSC osteogenic differentiation [46,47]. When undifferentiated SDC-1 KD cultures were differentiated, these cultures appeared to have enhanced osteogenic differentiation, with increased von Kossa and Alizarin Red staining. This indicates the undifferentiated cultures were primed for osteogenic differentiation following SDC-1 KD, despite the pro-adipogenic phenotype and demonstrates SDC-1 as a key factor in the osteo-adipogenic balance.

SDC-1 KD was then performed early in hMSC differentiation (on Day 2 of differentiation). The hMSC SDC-1 KD_AD_ cultures showed enhanced differentiation demonstrated by the increased Oil Red O staining, while the SDC-1 KD_OS_ cultures demonstrated impaired maturation, indicated by reduced Alizarin Red staining. This was further supported by the observed downregulation of the osteogenic marker COL1A1, as well as upregulation of the adipogenic marker PPARγ in the SDC-1 KD_AD_ cultures. None of the pathways examined showed significant differences between CTRL_AD/OS_ and SDC-1 KD_AD/OS_ conditions, in contrast to the effect on BMP2 and SHH signalling observed in undifferentiated hMSC SDC-1 KD cultures. This suggests BMP2/SHH may be the key initial pathway in this osteo-adipo balance interacting with SDC-1, but as hMSC cultures differentiate, a different pathway is activated or used. SDC-1 can also interact with other pathways, such as the epidermal growth factor (EGF) signalling pathway [48] that is also reported to regulate both adipogenesis and osteogenesis [3]; however, this pathway was not examined in this study. Comparison of SDC-1 KD cultures before and after initiation of lineage differentiation demonstrated a more profound effect after differentiation had commenced. This further suggests SDC-1 facilitates the reciprocity between osteogenic and adipogenic lineages in a stage-dependent manner and functions during the early induction stage of hMSC differentiation.

The ECM is a highly dynamic component of the cell microenvironment and controlled remodelling regulates a range of cellular functions including proliferation and differentiation [49]. As expected, following alteration to the in vitro niche through SDC-1 KD, active remodelling was evident through changes in HS GAG chain biosynthesis and modification enzyme expression, suggesting localised refinement of HS binding specificity in response to niche modification. Significant downregulation of HS chain initiation enzyme expression was observed in both SDC-1 KD_AD/OS_ cultures, suggesting a lack of HS chain initiation and elongation. The NDST1 enzyme removes *N*-acetyl groups to enable the addition of sulfate groups on HS chains and determines the overall GAG design, as all subsequent HS modifications rely on the presence of *N*-sulfated residues [50]. NDST1 gene expression was found to be significantly increased in hMSC SDC-1 KD_AD_ cultures, with a non-significant upregulation observed for NDST2. This correlated with a study by Forsberg et al. (2012) in which the authors demonstrated NDSTs are necessary for mouse embryonic stem cell adipogenesis but, that they are no essential for osteogenesis [51]. Furthermore, while 6-*O* sulfatases SULF1 and SULF2 expression remained unchanged in hMSC SDC-1 KD_AD/OS_ cultures, the significant upregulation of HS6ST1 gene expression observed in the hMSC SDC-1 KD_AD_ cultures indicates preservation and/or increased 6-*O*-sulfation. SULF1 has been reported to be downregulated during bone marrow hMSC adipogenesis [52] with both SULF1 and SULF2 reported to play key roles in osteogenesis [53], suggesting the maintenance of 6-*O*-sulfation is required for adipogenesis. This supports our observations of increased 6-*O*-sulfation in SDC-1 KD_AD_ cultures promoting enhanced adipogenesis.

Stem cell lineage commitment is determined by several factors. Apart from molecular controls, other elements including mechanical cues, chemical compounds, epigenetic regulators and biological factors such as ageing and metabolism ultimately contribute to cell fate [1,3,54]. McBeath and colleagues (2014) demonstrated hMSCs at lower densities are able to flatten and spread in order to undergo osteogenesis. In contrast, at a higher cell density, hMSCs become round, allowing greater lipid storage in the resulting adipocytes [3]. Additionally, it has been reported that seeding of hMSCs at a lower density inhibits adipogenesis with a high seeding density preventing osteogenesis [55], demonstrating cell density as a significant factor in hMSC lineage fate. The increased osteoblast formation observed in our osteogenic cultures with SDC-1 KD prior to differentiation was likely influenced by a decreased cell density in the cultures. The undifferentiated hMSC SDC-1 KD cultures had half the cell number/cell density of CTRL cultures prior to differentiation, matching the findings by McBeath et al. (2014), where cells switched to favouring osteogenesis as a result of culture density. In SDC-1 KD_OS/AD_ cultures, the SDC-1 KD_AD_ culture had significantly higher cell number/density, likely favouring adipogenesis. SDC-1 is known, largely in the context of cancer metastasis [42,56,57] to regulate cell proliferation with increased SDC-1 expression associated with higher cell proliferation. However, in the KD_AD/OS_ experiments, osteogenic cultures showed that despite similar cell numbers in both CTRL_OS_ and SDC-1 KD_OS_ conditions, osteogenesis was still impaired. This indicates SDC-1 KD impaired osteogenesis, and while SDC-1 control of cell proliferation is likely a contributing factor to lineage differentiation, there are other undetermined molecular events underpinning SDC-1 activity in the hMSC osteo-adipogenic balance that require further investigation.

## 4. Materials and Methods

### 4.1. Cell Culture and Niche Modification

Two commercially available primary hMSC populations (Lonza, Walkersville, MD, USA) were used in this study. They have previously been expanded and characterised in vitro and shown to be capable of osteogenic and adipogenic differentiation [58]. Undifferentiated hMSCs were plated at passage (P+) 7 in mesenchymal stem cell growth medium (MSCGM™), containing mesenchymal stem cell basal medium (MSCBM™), mesenchymal cell growth supplement (MCGS), L-glutamine, and gentamycin/ampicillin (GA-1000) SingleQuots™ (Lonza, Walkersville, MD, USA). Cells were plated at 5.0 × 10^3^ cells/cm^2^ in 24-well plates (Corning) in technical triplicates with cultures allowed to reach 50%–60% confluence prior to the introduction of Accell™ SMARTpool Human siRNAs (Dharmacon, Lafayette, CO, USA). The SMARTpool consisted of a set of four siRNA transcripts for specificity (see Table 1 for siRNA sequences) targeted for (SDC-1) knockdown (KD) in an Accell delivery media mix with 1 µM final siRNA concentration. Cells were incubated with either no siRNAs (untreated), non-targeting siRNAs or SDC-1-targeting siRNAs for 72 h or 96 h at 37 °C in 5% CO_2_ humidified atmosphere, after which cells were harvested for RNA and protein, respectively. Cell counts and viability were examined by Trypan Blue exclusion with an automated cell counter (Bio-Rad, Hercules, CA, USA) and via haemocytometer at the completion of the KD experiment. Technical triplicates were pooled prior to cell counts.

### 4.2. Adipogenic Differentiation and SDC-1 Knockdown

hMSC populations (*n* = 2) were plated at 2.1 × 10^4^ cells/cm^2^ onto 24-well plates in MSCGM™ in technical triplicates. The adipogenesis protocol was provided by the manufacturer (Lonza, Walkersville, MD, USA, Document #AA-2501-16 07/11). Once cells reached 90%-95% confluency, adipogenesis was induced by replacing the MSCGM™ with Adipogenesis Induction Medium (AIM) supplemented with SingleQuots™ of h-insulin (recombinant), L-glutamine, MCGS, dexamethasone, indomethacin, IBMX (3-isobutyl-l-methyl-xanthine) and GA-1000 (Lonza, Walkersville, MD, USA). After one day of adipogenesis, SDC-1 knockdown (SDC-1 KD_AD_) was induced by introducing siRNA in low serum (2%) AIM, along with no siRNA control (Untreated, UT_AD_) and non-targeting siRNA control (scrambled; SCR_AD_). hMSC adipogenic cultures were incubated with siRNA for 96 h and the medium replaced with Adipogenesis Maintenance Medium (AMM) consisting of SingleQuots™ of h-insulin (recombinant), L-glutamine, MCGS and GA-1000 (Lonza, Walkersville, MD, USA) for 2 days. hMSC cultures then underwent two more cycles of AIM (3 days) and AMM (2 days) followed by maintenance in AMM for 7 days prior to termination of differentiation (22 days in total). hMSC adipogenic cultures under SDC-1 knockdown conditions were harvested for RNA and protein, as well as stained for Oil Red O.

### 4.3. Oil Red O Staining

To examine the extent of adipogenesis, terminal adipogenic cultures were stained with Oil Red O for detection of lipid droplet accumulation, a characteristic of adipocytes [59]. hMSC adipogenic cultures were first fixed in 4% paraformaldehyde (PFA) at room temperature for 10 min. Stock Oil Red O (Sigma-Aldrich, Saint Louis, MO, USA) was prepared by dissolving 0.15 g of Oil Red O with 50 mL isopropanol and filtered with Minisart® syringe filters (0.2 µm; Sartorius, Göttingen, Germany). Working Oil Red O stain was prepared by mixing 4 parts Oil Red O stock with 3 parts Milli-Q water and dispensed onto hMSC cultures. Cells were incubated at room temperature for 15 min, rinsed and microscopic images of stained adipogenic cultures were obtained using a Nikon Eclipse Ts2 microscope at 10X magnification.

### 4.4. Osteogenic Differentiation and SDC-1 Knockdown

The osteogenesis protocol was provided by the manufacturer (Lonza, Walkersville, MD, USA, Document #AA-2501-16 07/11). hMSC populations (*n* = 2) were plated at 2.1 × 10^4^ cells/cm^2^ onto 24-well plates in MSCGM™ in technical triplicates. Once cells were attached, osteogenesis was induced by replacing the MSCGM™ with Osteogenic Induction Medium (OIM; Lonza, Walkersville, MD, USA), composed of hMSC Differentiation Basal Medium–Osteogenic with SingleQuots™ of dexamethasone, L-glutamine, ascorbate, penicillin/streptomycin, mesenchymal stem cell growth supplements, and β-glycerophosphate (Lonza, Walkersville, MD, USA). On Day 2 of differentiation, SDC-1 knockdown (SDC-1 KD_OS_) was induced by introducing siRNA in low serum (2%) OIM, along with no siRNA control (UT_OS_) and non-targeting siRNA control (SCR_OS_). Media change with OIM after 96 h of incubation with siRNA was performed to remove siRNA, and hMSCs were then cultured in supplemented OIM up to 21 days in total with medium replaced every 3–4 days. Technical triplicates were pooled for RNA and protein collection and the SDC-1 KD osteogenic cultures were also stained by Alizarin Red and von Kossa.

### 4.5. Alizarin Red and von Kossa Staining

To observe calcification and mineralisation of the hMSC SDC-1 KD osteogenic cultures, Alizarin Red staining was performed to detect calcium deposits. Mineralisation was observed through von Kossa staining, which detects anions of calcium salts (sulfates, phosphates, carbonates) in the osteogenic differentiation cultures. Both protocols were undertaken following the manufacturer’s instructions. Briefly, cells were fixed in 4% paraformaldehyde (PFA) and stained with either 0.1% Alizarin Red solution (Sigma-Aldrich, Saint Louis, MO, USA), adjusted to pH 4.3, or silver nitrate solution (Merck Millipore, Burlington, MA, USA) for 20 min under UV exposure for the von Kossa stain. For von Kossa staining, this was followed by addition of a sodium thiosulfate solution to remove un-reacted silver nitrate and cells were counterstained with a nuclear fast red-aluminium sulfate solution 0.1% (Merck Millipore, Burlington, MA, USA). Microscopic images of stained osteogenic cultures were obtained using a Nikon Eclipse Ts2 microscope at 10 X magnification.

### 4.6. Quantitation of hMSC Lineage Differentiated Stained Cultures

Digital images of stained cultures (Oil Red O, Alizarin Red and von Kossa) were obtained on a flatbed scanner. Quantitation of optical density (OD) of hMSC lineage specific stained cultures was conducted using the whole well and the Image J software (version 1.52q, NIH). First, OD calibration was performed by measuring the grey value of a Kodak No. 3 Calibrated Step Tablet (21 steps) with known OD values ranging from 0.05 to 3.05 OD. The calibration curve was generated using the ‘Rodbard’ function. Stained images were converted to 8-bit greyscale and OD of each culture condition was quantitated by subtracting background OD from the overall well OD. To account for the differences in cell number between culture conditions, OD was normalised to cell numbers and the results presented as OD ± standard deviation (SD).

### 4.7. RNA Isolation and cDNA Synthesis

For RNA harvest, technical triplicates of undifferentiated SDC-1 KD, SDC-1 KD_AD_ and SDC-1 KD_OS_ cultures of each hMSC population (*n* = 2) were pooled and homogenised using TRIzol reagent (Invitrogen, Carlsbad, CA, USA) and stored at -80 °C for a minimum of 24 h prior to extraction. RNA isolation was carried out using Direct-zol™ RNA MiniPrep kit (Zymo Research Corp., Irvine, CA, USA) with in-column DNase I (300 U/rxn) treatment. For cDNA synthesis, 150 ng of RNA was primed with 3 µg Random Primers (Invitrogen, Carlsbad, CA, USA) and 0.8 mM dNTP (New England BioLabs, Ipswich, MA, USA) in a total reaction volume of 25 µL. Samples were incubated at 65 °C for 10 min followed by 5 min at 4 °C. Following this, an additional 25 μL of cDNA synthesis mix (10X RT buffer, 4 mM MgCl_2_, 10 mM DTT, 40 U RNaseOUT and 200 U SuperScript III reverse transcriptase) was added to each reaction resulting in a total reaction volume of 50 µL and samples were incubated at 25 °C × 10 min, 50 °C × 50 min, 85 °C × 5 min.

### 4.8. Quantitative Real-Time PCR

For differential gene expression analysis, 120 ng of cDNA was amplified with SYBR^®^ Green PCR Master Mix (Promega, Madison, WI, USA), 100 µM each of primer pairs and 30 nM ROX Passive Reference Dye (Promega, Madison, WI, USA) in a 10 µL final reaction volume. PCR conditions were as followed: 50 °C × 2 min, 95 °C × 3 min followed by 50 cycles at 95 °C × 3 s, 60 °C × 30 s. Gene expression was normalised against the expression of endogenous *18S* ribosomal RNA and calculated using the 2^−ΔCT^ method. Q-PCR for each of the two hMSC populations (technical triplicates pooled) was performed in quadruplicate, giving a total of 8 replicates per gene examined. Primer sequences for HSPG core proteins (SDCs and GPCs), HS biosynthesis and modification enzymes, and lineage specific markers can be found in our previous studies [58,60], pathway primers used can be found in Table 2. All data are presented as mean ± standard error of mean (SEM) unless otherwise stated. Statistical significance was set at * *p* < 0.05, ** *p* < 0.01, *** *p* < 0.001, as determined by Student’s *t*-test.

### 4.9. Western Blotting

Technical triplicates of undifferentiated SDC-1 KD, SDC-1 KD_AD_ and SDC-1 KD_OS_ cultures of each hMSC population (*n* = 2) were pooled prior to protein collection and total protein was extracted using RUNX protein-lysis buffer (20mM HEPES, 25% glycerol, 1.5 mM MgCl_2_, 420 mM NaCl, 0.5 mM DTT, 0.2 mM EDTA, 0.5% Igepal CA-630, 0.2 mM sodium orthovanadate (Na_3_VO_4_) and 1 mM phenylmethylsulfonyl fluoride (PMSF) containing protease and phosphatase inhibitors at 1:100 dilution). The two hMSC populations were pooled and samples quantitated using the Pierce^®^ BCA Protein Assay Kit (Thermo Scientific, Rockford, IL, USA). Approximately 20–30 μg of protein was separated by SDS-PAGE using 12% TGX^TM^ FastCast^TM^ Acrylamide gels (Bio-Rad, Gladesville, NSW, Australia) and transferred onto 0.2 μm PVDF membranes (EMD Millipore Corp., Billerica, MA, USA) using the Bio-Rad Transblot turbo system. The membrane was blocked with 5% skim milk in TBST (Tris-buffered saline + 0.1% Tween-20) followed by addition of primary antibodies diluted in 5% BSA in TBST and incubated overnight at 4 °C. Primary antibodies used were: anti-PPARγ (sc-7273, Santa Cruz Biotechnology, Santa Cruz, CA, USA, 1:200), anti-COL1A1 (sc-28657, Santa Cruz Biotechnology, Santa Cruz, CA, USA, 1:200), anti-syndecan-1 (ab34164, Abcam, Cambridge, MA, USA, 1:200), anti-Δ3G10 (H1890-75, US Biological, Salem, MA, USA, 1:1000) with GAPDH (21185, Cell Signaling, Danvers, MA, USA, 1:1000) used as the loading control. Primary antibodies were removed the next day, membranes were washed in TBST and then incubated with HRP-conjugated secondary antibodies (anti-Mouse IgG, #7076 and anti-Rabbit IgG, #7074, both from Cell Signaling, Danvers, MA, USA, diluted 1:3000 in 5% BSA in TBST) for 2 h at room temperature. Target protein detection was performed with enhanced chemiluminescence (Clarity ECL, Bio-Rad, Gladesville, NSW, Australia) using Fusion FX Spectra chemiluminescence system (Vilber Lourmat, Fisher Biotec, Wembley, WA, Australia) and quantitated using Image J software (version 1.52q, NIH).

### 4.10. Ethical Statement

Human mesenchymal stem cell (hMSC) populations (*n* = 2) used in this study was commercially obtained from Lonza, Walkersville, MD, USA (Catalog #: PT-2501). Approval for use was granted by the QUT Human Research Ethics Committee (UHREC), Approval #: 1800000179.

## 5. Conclusions

Current evidence suggests SDC-1 is a key facilitator of the hMSC osteo-adipogenic balance, where adipogenesis is favoured following reduced SDC-1 and reciprocally osteogenesis is impaired. However, SDC-1 does not function in isolation, and how to specifically use the role of SDC-1 and other HSPGs for therapeutic applications warrants further investigation. This will likely include the examination of direct and indirect interactions of SDC-1 and other HSPGs with signalling ligands and key osteogenic and adipogenic transcription factors. The current study supports SDC-1 modulation of the osteo-adipogenic balance. Knockdown of SDC-1 impaired osteogenesis while enhancing adipogenesis, implicating SDC-1 as a key biomarker of hMSC differentiation. Understanding the factors controlling hMSC fate has significant implications for a wide range of human health challenges, from osteoporosis to tissue regeneration. MSC therapy provides promise in therapeutic approaches such as tissue regeneration and repair in a wide variety of diseases, particularly in ageing populations. A better understanding of the molecular mechanisms underlying MSC expansion and lineage specification will allow the identification of novel markers, to provide greater control over the use of hMSCs in these applications.

## Figures and Tables

**Figure 1 ijms-21-03884-f001:**
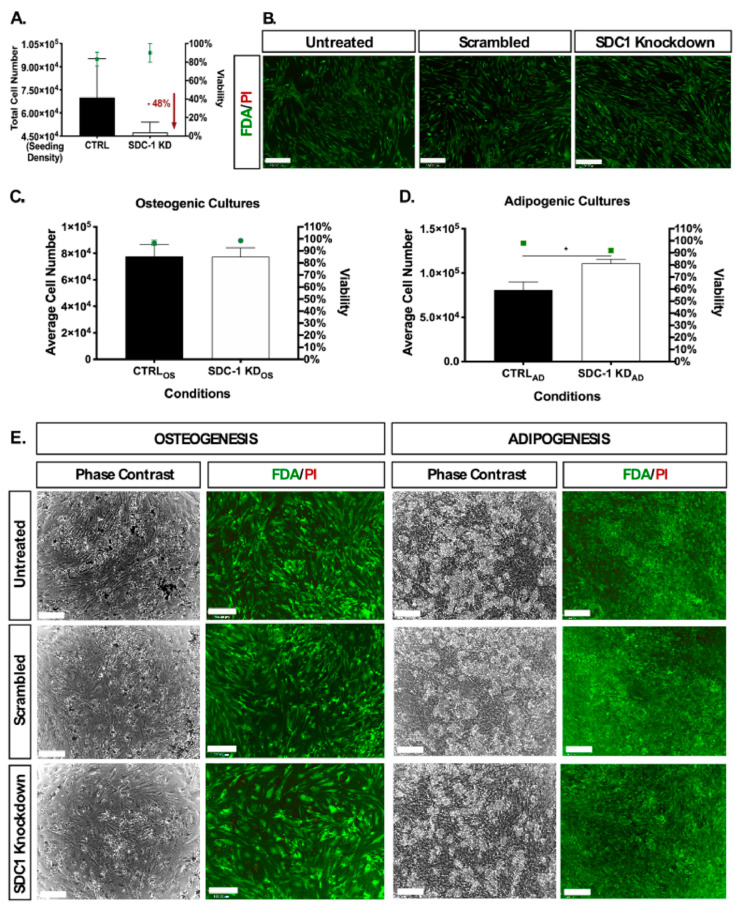
Cell count and viability of human mesenchymal stem cell (hMSC) populations following syndecan-1 (SDC-1) knockdown (KD) in undifferentiated and differentiated cultures. (**A**) Total cell number and viability of undifferentiated hMSC (*n* = 2) cultures following SDC-1 KD compared to control (CTRL) cultures (combined untreated and non-targeting/scrambled siRNA data), following 72 h of incubation with SDC-1 specific siRNA. SDC-1 KD resulted in a 48% decrease in total cell number when compared to CTRL cultures with high viability, CTRL (83%) and SDC-1 KD (90%), maintained in all cultures. Left Y-axis starts at initial seeding density = 4.5 × 10^4^ cells/condition and data is presented as a single value after triplicate data was pooled. (**B**) FDA/PI (green = live/red = dead) analysis correlates with high viability of undifferentiated hMSC SDC-1 KD cultures with high FDA staining observed at 72 h. (**C**) Average cell number and viability of terminally differentiated hMSC osteogenic cultures +/− SDC-1 KD (CTRL_OS_ and SDC-1 KD_OS_) recorded following 21 days of differentiation. (**D**) Average cell number and viability of terminally differentiated hMSC adipogenic cultures +/− SDC-1 KD (CTRL_AD_ and SDC-1 KD_AD_) recorded after 22 days of differentiation. Graph bars = cell number, green graph data points = viability, all error bars are presented as SEM. Statistical differences in cell numbers were determined by Student’s *T* test and significance denoted by * *p* < 0.05. (**E**) FDA/PI cell viability analysis of terminally differentiated osteogenic and adipogenic cultures in untreated (UT_AD/OS_), scrambled (SCR_AD/OS_) or SDC-1 KD_AD/OS_ cultures. All cell images were taken at 4X magnification, scale bar = 200 µm.

**Figure 2 ijms-21-03884-f002:**
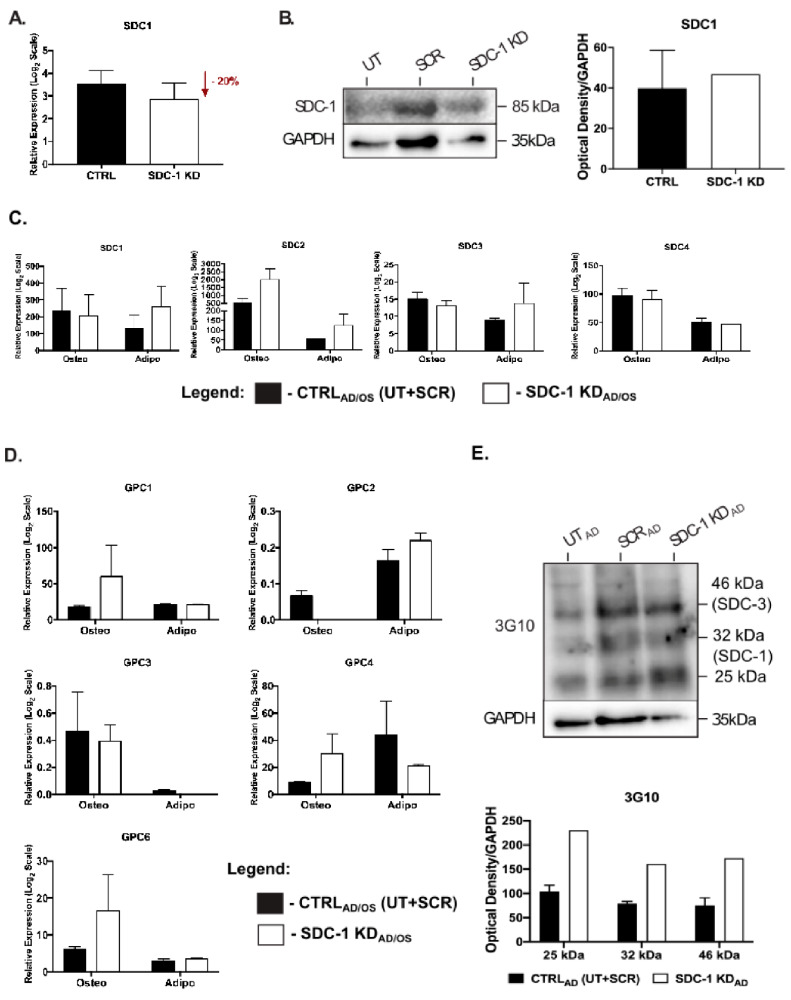
Change in expression of heparan sulfate proteoglycan (HSPG) core proteins following SDC-1 KD in hMSC undifferentiated and differentiated cultures. SDC-1 KD resulted in (**A**) a 20% reduction in SDC-1 gene expression in undifferentiated hMSCs when compared to control (CTRL; Untreated (UT) + Scrambled (SCR)) cultures; and (**B**) an 18% increase in protein expression identified by Western blot analysis of SDC-1 core protein expression via anti-syndecan-1 antibody (ab34164, Abcam). SDC-1 protein was assessed in UT, SCR and SDC-1 KD samples after 96 h of incubation of SDC-1 specific siRNA, with GAPDH as the loading control. Gene expression analysis by Q-PCR of (**C**) Syndecan-1-4 (SDC1-4) and (**D**) Glypican-1-4, and -6 (GPC-1-4, -6) core protein gene expression in differentiated hMSC CTRL_AD/OS_ and SDC-1 KD _AD/OS_ cultures. Error bars represent SEM. (**E**) Western blot analysis of HS 3G10 epitope (H1890-75, US Biological) in hMSC UT_AD_, SCR_AD_ and SDC-1 KD_AD_ samples. Optical density of protein bands quantitated using ImageJ (NIH) and target signals were normalised to the loading control GAPDH.

**Figure 3 ijms-21-03884-f003:**
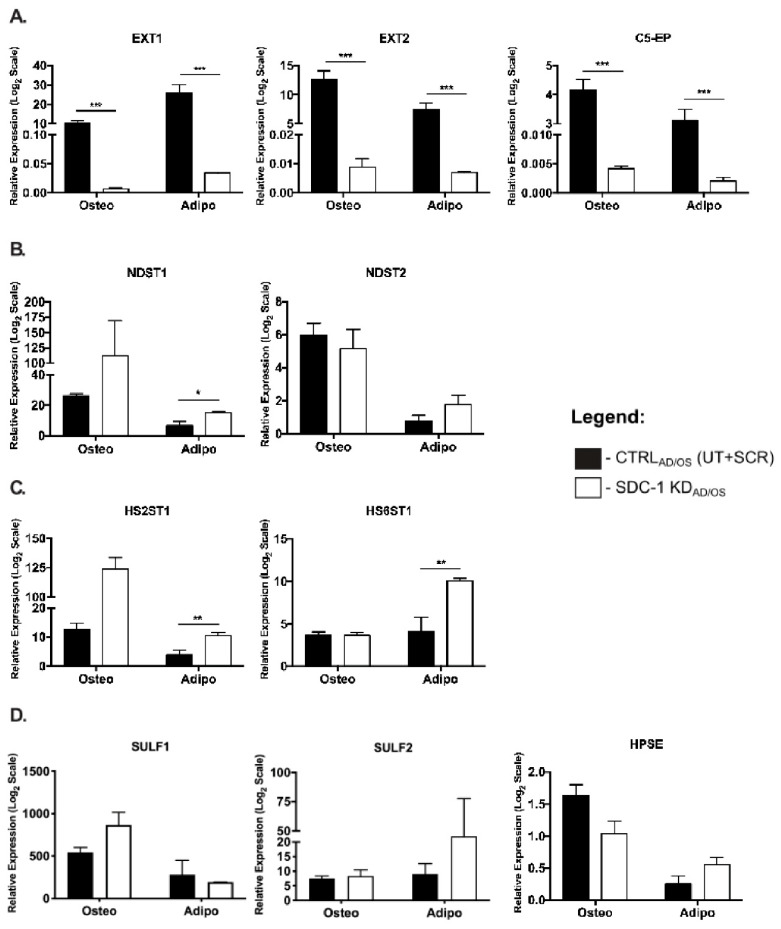
Gene expression level changes of heparan sulfate biosynthesis and modification enzymes in CTRL_AD/OS_ and SDC-1 KD_AD/OS_ hMSC differentiated cultures. Gene expression levels of (**A**) exostosins 1-2 (EXT1-2) and C-5 epimerase (C5-EP), (**B**) *N*-deacetylase/*N*-sulfotransferases 1-2 (NDST1-2), (**C**) heparan sulfate 2-*O*-sulfotransferase 1 (HS2ST1) and heparan sulfate 6-*O*-sulfotransferase 1 (HS6ST1), and (**D**) sulfatases 1-2 (SULF1-2) and heparanase (HPSE) in differentiated hMSC CTRL_AD/OS_ and SDC-1 KD_AD/OS_ cultures. Gene expression analysis was performed with terminally differentiated hMSC cultures, in which RNA was collected after 21 days (osteogenesis) and 22 days (adipogenesis) of differentiation. Significant difference in expression levels were analysed by Student’s T test, significance was set at * *p* < 0.05, ** *p* < 0.01 and *** *p* < 0.001. Error bars represent SEM.

**Figure 4 ijms-21-03884-f004:**
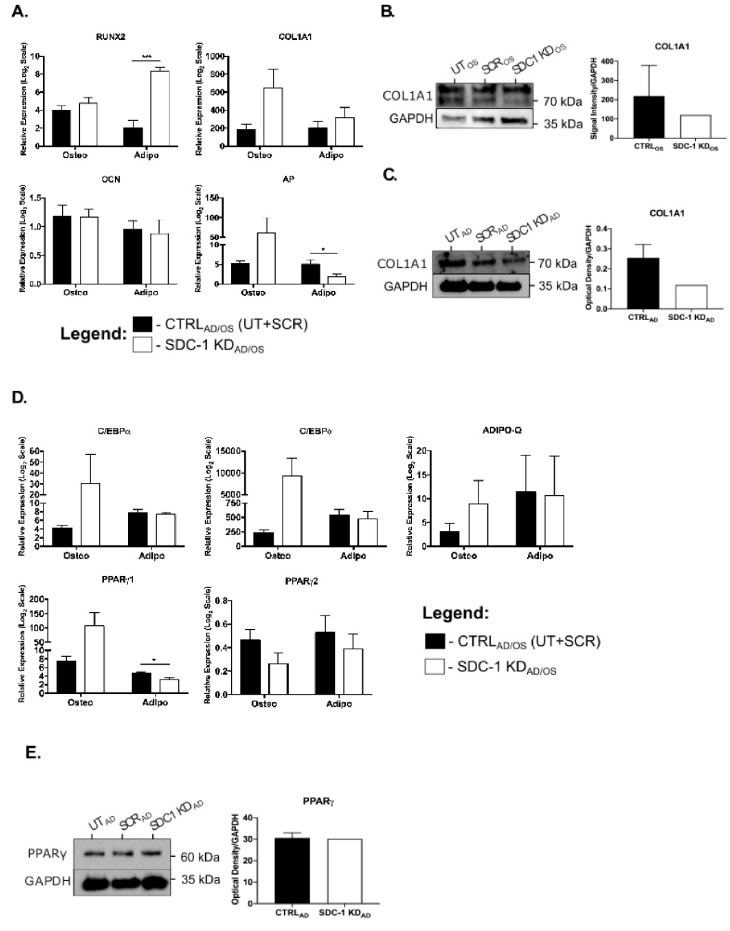
Osteogenic and adipogenic lineage marker expression changes in hMSC SDC-1 KD_AD/OS_ lineage differentiation cultures. Analysis of lineage specific markers was performed with terminally differentiated hMSC cultures, in which RNA and protein was collected after 21 days (osteogenesis) and 22 days (adipogenesis) of differentiation. (**A**) Gene expression changes of osteogenic markers runt-related transcription factor 2 (RUNX2), collagen, type I, alpha 1 (COL1A1), osteocalcin (OCN) and alkaline phosphatase (AP) in hMSC CTRL_AD/OS_ (UT+SCR) and SDC-1 KD_AD/OS_ cultures. Western blot analysis and optical density quantitation of osteogenic marker COL1A1 in UT_AD/OS_, SCR_AD/OS_ and SDC-1 KD_AD/OS_ conditions in (**B**) hMSC osteogenic cultures and (**C**) hMSC adipogenic cultures. (**D**) Gene expression analysis of adipogenic markers CCAAT/enhancer binding protein alpha (C/EBPα), CCAAT/enhancer binding protein delta (C/EBPδ), peroxisome proliferator-activated receptor gamma 1 (PPARγ1) and 2 (PPARγ2), and adiponectin (ADIPO-Q) in hMSC CTRL_AD/OS_ (UT+SCR) and SDC-1 KD_AD/OS_ cultures. (**E**) Western blot analysis and optical density quantitation of adipogenic marker PPARγ in UT_AD_, SCR_AD_ and SDC-1 KD_AD_ conditions of hMSC adipogenic cultures. Significant differences in gene expression were determined by Student’s T test. Significance level was set at * *p* < 0.05 and *** *p* < 0.001. Error bars represent SEM. Quantitation of optical density of Western blots was performed using ImageJ (NIH), error bar represent SD.

**Figure 5 ijms-21-03884-f005:**
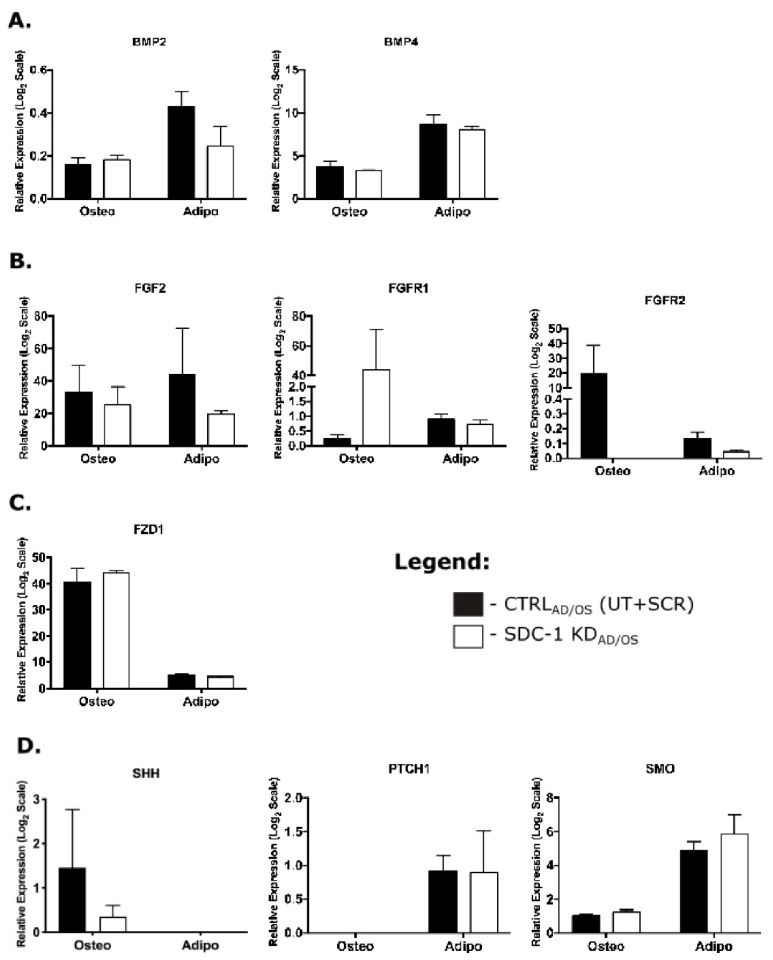
Gene expression changes in common signalling pathways implicated in osteogenic and adipogenic lineages. Gene expression analysis was performed on terminally differentiated hMSC cultures, in which RNA was collected after 21 days (osteogenesis) and 22 days (adipogenesis) of differentiation. Changes in gene expression in CTRL_AD/OS_ and SDC-1 KD_AD/OS_ cultures detected by Q-PCR of (**A**) Bone morphogenetic protein (BMP) signalling including the ligands BMP2 and BMP4, and BMP receptors IA (BMPR-IA) and IB (BMPR-IB). (**B**) Fibroblast growth factor (FGF) signalling with ligand FGF2 and the receptors FGFR1 and FGFR2. (**C**) Wnt receptor frizzled (FZD1). (**D**) Sonic hedgehog signalling (SHH) including the ligand SHH, the canonical receptor Patched (PTCH1) and the G protein-coupled receptor-like receptor Smoothened (SMO). Error bars represent SEM.

**Figure 6 ijms-21-03884-f006:**
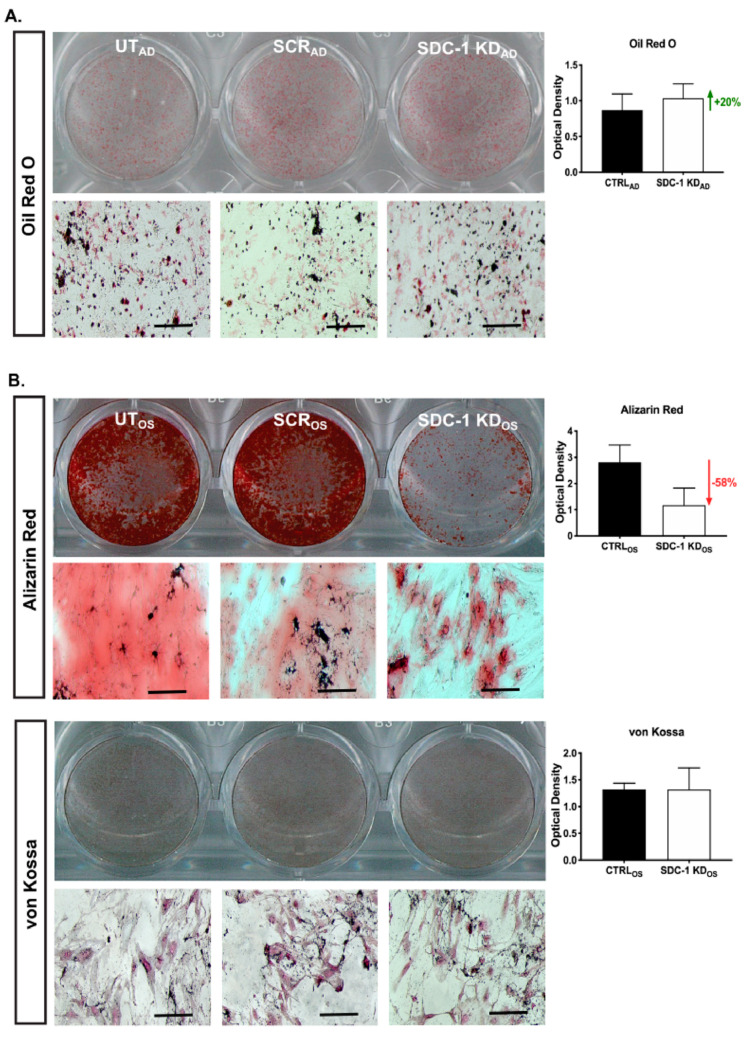
Oil Red O, Alizarin Red and von Kossa staining of hMSC SDC-1 KD_AD/OS_ cultures following in vitro adipogenic (22 days) and osteogenic (21 days) differentiation. (**A**) hMSC SDC-1 KD_AD_ cultures stained with Oil Red O. Plate image shows representative untreated (UT_AD_), scrambled (SCR_AD_) and SDC-1 knockdown (SDC-1 KD_AD_) conditions, with corresponding brightfield images (bottom panel). Quantitation of staining intensity of control (CTRL_AD_: UT_AD_ and SCR_AD_ combined) conditions and SDC-1 KD_AD_ conditions showed a 20% increase in Oil Red O staining in SDC-1 KD_AD_ cultures. (**B**) hMSC SDC-1 KD_OS_ cultures: UT_OS_, SCR_OS_ and SDC-1 KD_OS_ cultures with corresponding brightfield images stained with Alizarin Red and von Kossa staining. Alizarin Red staining of calcium deposits showed a 58% decrease in the SDC-1 KD_OS_ cultures compared to CTRL_OS_ cultures. The von Kossa stain targets anions of calcium salt, with equal staining observed between CTRL_OS_ and SDC-1 KD_OS_ cultures. Stained images were acquired using a flatbed scanner and signals quantitated using ImageJ (NIH). Microscopic brightfield images were taken using a Nikon Eclipse Ts2 microscope at 10× magnification, scale bar = 200 µm. Optical density of stain signals was normalised to cell number due to the significant differences observed between CTRL_AD/OS_ and SDC-1 KD_AD/OS_ cultures (as presented in Figure 1C,D).

**Table 1 ijms-21-03884-t001:** siRNA sequences.

Accell siRNA		Sequences
**Non-targeting** (Scrambled; SCR)	siRNA-1	UGGUUUACAUGUCGACUAA
	siRNA-2	UGGUUUACAUGUUUUCUGA
	siRNA-3	UGUUUACAUGUUUUCCUA
	siRNA-4	UGGUUUACAUGUUGUGUGA
**SDC-1**	siRNA-1	UGCUUAUUUGACAACGUUU
	siRNA-2	CUCUAGUUCUUUGUUCAUA
	siRNA-3	GUGUUGUCUCUUGAGUUUG
	siRNA-4	GGUUCAGCCAAGGUUUUAU

**Table 2 ijms-21-03884-t002:** Primer sequences for Q-PCR.

Gene Name	Symbol		Sequence	Amplicon (bp)	Ref Seq.
**BMP signalling**					
Bone morphogenetic protein 2	BMP2	F	ACCCGCTGTCTTCTAGCGT	180	NM_001200
		R	TTTCAGGCCGAACATGCTGAG		
Bone morphogenetic protein 4	BMP4	F	ATGATTCCTGGTAACCGAATGC	93	NM_001202
		R	CCCCGTCTCAGGTATCAAACT		
Bone morphogenetic protein receptor type 1A	BMPR-IA	F	TGAAATCAGACTCCGACCAGA	150	NM_004329.2
		R	TGGCAAAGCAATGTCCATTAGTT		
Bone morphogenetic protein receptor type 1B	BMPR-IB	F	CTTTTGCGAAGTGCAGGAAAAT	130	NM_001256793.1
		R	TGTTGACTGAGTCTTCTGGACAA		
**FGF signalling**					
Fibroblast growth factor 2	FGF2	F	AAAAACGGGGGCTTCTTCCT	86	NM_002006.4
		R	TGTAGCTTGATGTGAGGGTCG		
Fibroblast growth factor receptor 1	FGFR1	F	CCGTATGTCCAGATCCTGAAGA	126	NM_001354368.1
		R	GATAGAGTTACCCGCCAAGCA		
Fibroblast growth factor receptor 2	FGFR2	F	TCAAGGTTCTCAAGCACTCGG	89	NM_022970.3
		R	ATATTCCCCAGCATCCGCCT		
Fibroblast growth factor receptor 3	FGFR3	F	CCCTACGTCACTGTACTCAAGACTG	80	NM_000142.5
		R	GTGACATTGTGCAAGGACAGAAC		
**Wnt signalling**					
Wnt family member 3A	Wnt3a	F	GTTGGGCCACAGTATTCCT	111	NM_033131.3
		R	GGGCATGATCTCCACGTAGT		
Frizzled-1	FZD1	F	ACCAACAGCAAACAAGGGGA	163	NM_003505.1
		R	GGAGCCTGCGAAAGAGAGTT		
**SHH signalling**					
Sonic hedgehog	SHH	F	TTATCCCCAATGTGGCCGAG	168	NM_000193.3
		R	TACACCTCTGAGTCATCAGCCT		
Patched-1	PTCH1	F	GGAGCAGATTTCCAAGGGGA	128	NM_001354918.1
		R	CCACAACCAAGAACTTGCCG		
Smoothened	SMO	F	TCAGCTGCCACTTCTACGAC	83	NM_005631.4
		R	CACATTGGCCTGACATAGCAC		

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
