# Peer review of "Syndecan-1 Facilitates the Human Mesenchymal Stem Cell Osteo-Adipogenic Balance"

_ijms, 2020, doi:10.3390/ijms21113884_

Round 1

Reviewer 1 Report

The manuscript  by Chieh Yu et al.  aims at demonstrating the role of  SDC-1 in hMSCs osteo-adipogenic fate.  

Experimental data are difficult to understand also because of incomplete figure legends;  the fuzzy text style is not very helpful  either.

Major criticisms:

  • The authors claim they disrupted the HSPG profile of primary hMSc cultures by knockdown of SDC-1 using RNA interference. 2 The authors observed only 20% gene expression downregulation: this is a very inefficient silencing.  Maybe the experimental conditions need to be improved. The protein expression level actually increased in the KD cells compared to untreated cells.
  • 4 shows osteogenic and adipogenic lineage marker  genes expression changes.  What is crucial to know for a correct interpretation of the data is: on which day of either differentiation gene expression evaluation was performed?
  • Similarly, in Fig.5, no information is reported about on which day of differentiation gene expression changes in mutually exclusive signaling pathways (adipogenic/osteogenic) were monitored.

In this reviewer’s opinion, the authors’conclusions are  not supported by solid and consistent data.

Author Response

The manuscript by Chieh Yu et al. aims at demonstrating the role of SDC-1 in hMSCs osteo-adipogenic fate.  

Experimental data are difficult to understand also because of incomplete figure legends; the fuzzy text style is not very helpful either.

We apologise to the Reviewer for these difficulties. We have now revised all the Figure legends (for both the main figures and supplementary figures) to ensure completeness and better clarity. Specifically, the timepoints (at 72 h or 96 h after knockdown, or at termination of differentiation cultures) of data presented have been clearly stated. We have also ensured all data figures have been provided in a .tiff file format with of a minimum 300 dpi resolution before insertion into the main text, this has also ensured that the text within the figures are also now clearer.  

Major criticisms:

  • The authors claim they disrupted the HSPG profile of primary hMSC cultures by knockdown of SDC-1 using RNA interference. The authors observed only 20% gene expression downregulation: this is a very inefficient silencing. Maybe the experimental conditions need to be improved. The protein expression level actually increased in the KD cells compared to untreated cells.

We thank the Reviewer and agree that the 20% downregulation of SDC-1 gene expression could be considered low transfection efficiency and that the SDC-1 protein expression level assessed by WB analysis appeared to be slightly increased in the SDC-1KD cultures. The slight increase in protein expression could be due to a number of reasons. In our manuscript we have proposed the protein level would likely have declined at a later timepoint, as we examined the protein level at one timepoint (96 h) after cells were incubated with SDC-1 specific siRNAs as outlined below:

Section 3, page 16, Lines 420-423:

As SDC-1 KD efficiency was only examined at one timepoint following KD, at 72 h for RNA expression and 96 h for protein expression, this likely explains why SDC-1 KD was only achieved at the transcript level and not also at the protein level, although a delayed downregulation of SDC1 protein may have occurred.

In addition, we have also now added additional reasoning for the slight increase in SDC‑1 protein level observed in the Discussion Section of the manuscript. Briefly, the shedding of SDC‑1 ectodomains from the cell surface into the extracellular compartment may account for this likely transient increase in SDC1 protein in the cultures. The additional text within the manuscript is outlined below:

Section 3, Page 16, Line 423-429:

Another possibility for this likely transient small increase in SDC1 protein is the shedding of SDC1 from the cell surface. In this process, the ectodomain of SDC-1 is subjected to proteolytic cleavage in response to physiological agents, releasing the protein into the extracellular matrix as a soluble ligand [42]. As Western blotting was performed using cell lysates, containing both cell surface-bound and extracellular matrix-localised heparan sulfate proteoglycans, while SDC‑1 was downregulated due to decreased transcription following KD, SDC-1 shedding may have contributed to the slight increase in total SDC-1 protein level observed at the 96 h timepoint.

In regard to the Reviewer’s query regarding the SDC-1 knockdown efficiency at the transcript level, the 20% reduction in SDC‑1 expression in the KD cultures is considered at the low end. However, this 20% reduction in transcript levels did result in significant changes in hMSC culture phenotype and lineage as outlined in the manuscript. To acknowledge this, we have now further elaborated on this in the manuscript and have provided specific evidence for the validity of our SDC-1 knockdown efficiency in the Discussion, as below:

Section 3, Page 16, Line 430-443:

SDC-1 is associated with proliferation through its interaction with growth factors and is often upregulated in cancer cells during tumour progression [42,43]. In this study, SDC‑1 KD resulted in a 20% decrease in SDC‑1 gene expression, which is considered to be a knockdown of low efficiency. Primary cells, including hMSCs, are generally more difficult to transfect than cell lines [45]. However, the 20% decrease in SDC-1 gene expression, resulted in a 48% reduction in cell number (Fig. 1A), in undifferentiated SDC-1 KD cultures when compared to undifferentiated CTRL cultures. A previous study in human multipotent adipose stem (hMADS) cells, which similar to hMSCs are derived from the mesenchyme, were observed to have an approximate 50% reduction in cell number following SDC-1 KD [32]. The decreased cell number observed following SDC-1 KD indicates an effect on SDC-1 function. Significant changes in the HSPG profile, including significant changes in core protein (Supplementary Fig. S1) and biosynthetic enzyme expression (Supplementary Fig. S2) also indicate the HSPG profile of hMSC cultures was altered following SDC‑1 KD. This is further supported by the observed downregulation of osteogenic markers at the gene and protein level (Supplementary Fig. S3A & S3B), and the significant upregulation of the adipogenic marker (ADIPO-Q; Supplementary Fig. S3C), suggesting an altered osteo-adipogenic balance in the hMSC SDC-1 KD cultures.

  • 4 shows osteogenic and adipogenic lineage marker genes expression changes. What is crucial to know for a correct interpretation of the data is: on which day of either differentiation gene expression evaluation was performed?

We thank the Reviewer for noting the omission of this crucial information. To clarify, in Figure 4, osteogenic and adipogenic linage markers were examined in terminally differentiated hMSC cultures. For osteogenesis, cultures were terminated on Day 21, and for adipogenesis, cultures were terminated on Day 22. This is in accordance with the differentiation protocol supplied by the manufacturer (Lonza, Document #AA-2501-16 07/11). Reference to this protocol and the number of days of differentiation is now stated in the Methods section of the revised manuscript, as outlined below:

Section 4.2, Page 17, Line 466-467:

“The adipogenesis differentiation protocol performed was provided by the manufacturer (Lonza, Document #AA-2501-16 07/11).”;

Section 4.2, Page 17, Line 476:

hMSC cultures then underwent two more cycles of AIM (3 days) and AMM (2 days) followed by maintenance in AMM for 7 days prior to termination of differentiation (22 days in total).”;

Section 4.4, Page 17, Line 489-490:

The osteogenesis protocol performed was provided by the manufacturer (Lonza, Document #AA-2501-16 07/11).”;

And as already included in the original manuscript, Section 4.4, Page 18, Line 500-502:

Media change with OIM after 96 h of incubation with siRNA was performed to remove siRNA and hMSCs were then cultured in supplemented OIM up to 21 days in total with medium replenished every 3-4 days.

Importantly, the Figure legend for Figure 4 have now been updated to also include this information, as follows:

Section 2.3, Page 10, Line 254-256:

Analysis of lineage specific markers was performed with terminally differentiated hMSC cultures, in which RNA and protein was collected after 21 days (osteogenesis) and 22 days (adipogenesis) of differentiation.

  • Similarly, in Fig. 5, no information is reported about on which day of differentiation gene expression changes in mutually exclusive signaling pathways (adipogenic/osteogenic) were monitored.

We thank the Reviewer again for bringing this point to our attention. For Figure 5, examination of gene expression changes in signaling pathways was also conducted in terminally differentiated hMSC cultures, therefore, on Day 21 for osteogenic cultures and on Day 22 for adipogenic cultures. The figure legend for Figure 5 has also now been updated to include information on the days of differentiation in which gene expression changes were monitored, as follows:

Section 2.4, Page 12, Lines 304-306:

Gene expression analysis was performed on terminally differentiated hMSC cultures, in which RNA was collected after 21 days (osteogenesis) and 22 days (adipogenesis) of differentiation.

We have also now specified the experimental timepoints in all of the Figure legends (main figures and supplementary figures) for better interpretation of the data presented in this study.

  • In this reviewer’s opinion, the authors’ conclusions are not supported by solid and consistent data.

We thank the Reviewer for this honest assessment. In an effort to better convey and summarise our findings, we have now significantly reviewed our Discussion and Conclusion sections, including the citation of additional references in support of our findings. Briefly, our study investigated the role of SDC-1 in the hMSC osteo-adipogenic balance. In addition, we have also addressed the validity of the SDC-1 KD efficiency our response to Comment 2 above. The knockdown of SDC-1 in both undifferentiated and differentiated hMSC cultures demonstrated a pro-adipogenic phenotype and enhanced adipogenesis, respectively, with osteogenesis impaired, as supported by culture phenotype (changes in lineage marker expression at the RNA and protein level, as presented in Fig. 4 and Supplementary Fig. S3), changes in HSPG profile (of SDC-1 and other HSPG core proteins, as well as HS biosynthetic enzymes, Fig. 2, Fig. 3, Supplementary Fig. S1, and Supplementary Fig. S2) and subsequent functionality of SDC-1 KD differentiated hMSC cultures (presented by Oil Red O, Alizarin Red and von Kossa staining and quantitation, Figure 6 and Supplementary Figure S6). Overall, our data supports SDC-1 KD promoting adipogenesis as the expense of osteogenesis. We have also now included in our Discussion, other reports supporting our conclusion:

Section 3, Page 16, Line 420-422

Specifically, SDC-1 gene expression was previously demonstrated to be downregulated during human adipocyte differentiation [32,41], whilst murine studies show SDC-1 was significantly upregulated in osteoprogenitors and the osteogenic differentiation condition [27,31].

Additionally, to make our Conclusion clearer, it now reads:

Section 5, Page 21, Line 624-628

Current evidence suggests SDC-1 is a key facilitator of hMSC osteo-adipogenic balance, where adipogenesis is favoured when SDC-1 is knocked down and reciprocally osteogenesis is impaired. However, SDC-1 does not function in isolation, and how to specifically utilise the role of SDC-1 and other HSPGs for therapeutic applications warrants further investigation. This will likely include the examination of direct and indirect interactions of HSPGs with signalling ligands and key osteogenic and adipogenic transcription factors.

We believe the data we have presented, and supported by other studies in the role of SDC-1, reflects our conclusion accurately.

Reviewer 2 Report

The paper titled 'Syndecan-1 Facilitates the Human Mesenchymal Stem Cell Adipo-Osteogenic Balance' is well designed and performed.

The present study is focused on the implication of SDC-1 28 as a facilitator of the hMSC adipo-osteogenic balance during early induction of differentiation.

Authors should provide for figure 6 the cells images  obtained by microscope to complete the panel.

Also authors should better discuss the hMSCs (e.g. doi: 10.3390/ijms20204987; doi: 10.3390/ijms21093242).

Author Response

  • The paper titled 'Syndecan-1 Facilitates the Human Mesenchymal Stem Cell Adipo-Osteogenic Balance' is well designed and performed. The present study is focused on the implication of SDC-1 as a facilitator of the hMSC adipo-osteogenic balance during early induction of differentiation.

We thank the reviewer for this feedback.

  • Authors should provide for figure 6 the cells images obtained by microscope to complete the panel.

We thank for the Reviewer for this helpful suggestion and have now provided additional microscopy images to Figure 6 to complete the existing panels. To acquire these images, differentiated hMSC SDC-1 KDAD/OS cultures were imaged using a Nikon Eclipse Ts2 microscope at 10X magnification on brightfield. We have updated the figure legend for Figure 6, and Sections 4.3 & 4.5 in the Methods to include this information.

  • Also authors should better discuss the hMSCs (e.g. doi: 10.3390/ijms20204987; doi: 10.3390/ijms21093242).

We again thank the Reviewer for this suggestion and for providing examples of studies demonstrating the importance of hMSCs as a cell source in tissue repair and regeneration. We have now expanded our Discussion on hMSCs in Section 3 of the manuscript. This section now provides more emphasis on the significance of our work and provides additional evidence of our current understanding of the adipo-osteogenic balance in the context of improving the use of hMSCs for use in understanding adipose and bone pathophysiology and their application to regenerative medicine, as follows:

Section 3, Page 14, Lines 355-357:

MSCs possess numerous properties such as their ease of isolation, high ex vivo expansion capability, homing ability, immunomodulatory properties and multipotency, making them an ideal cell type for tissue repair and regeneration [37,38]”;

and

Section 3, Page 15, Lines 431-434:

A better understanding of the inverse relationship of MSC adipocyte and osteoblast lineages has significant implications, from insights into pathophysiological conditions such as obesity and osteoporosis, to the development of treatments for these disorders and MSC engineering for bone tissue repair [16,39]”.

Round 2

Reviewer 1 Report

I acknowledge  the author’s efforts to fulfil the reviewer’s queries. I believe that the manuscript is more understandable now. The revised Discussion and Conclusion sections, strengthened by additional references, suggest new “food for thought” and propose interesting further investigations.

Moderate English editing required

Author Response

We have now fully revised and edited the language and English in the manuscript and believe it is now ready for publication.